# The Relationship between Executive Functions and Dance Classes in Preschool Age Children

Elena Chichinina [1,2,*][ID], Daria Bukhalenkova [1,2][ID], Alla Tvardovskaya [3][ID], Yury Semyonov [4], Margarita Gavrilova [1,2][ID] and Olga Almazova [5]

1 Department of Educational Psychology and Pedagogy, Faculty of Psychology, Lomonosov Moscow State University, Moscow 125009, Russia
2 Laboratory of Childhood Psychology and Digital Socialization, Psychological Institute, Russian Academy of Education, Moscow 125009, Russia
3 Institute of Psychology and Education, Kazan Federal University, Kazan 420021, Russia
4 Scientific and Educational Center of the State Institution Academy of Sciences of the Republic of Sakha (Yakutia), Yakutsk 677007, Russia
5 Department of Developmental Psychology, Faculty of Psychology, Lomonosov Moscow State University, Moscow 125009, Russia
* Correspondence: alchichini@gmail.com

**Abstract:** The development of executive functions is of the utmost importance for academic success at school and the social adaptation of children. Dance class attendance is one of the factors promoting the development of these functions in children. This study was aimed to explore the relationship between extra dance class attendance and executive functions in preschool age children. The executive function level was assessed using NEPSY-II subtests "Sentences Repetition", "Memory for Designs", "Inhibition", "Statue", and "Dimensional Change Card Sort". The data on extra dance classes were collected by means of a questionnaire for parents. In the study, 86 typically developing 5–6-year-old children participated. No statistically significant differences in executive functions' levels were discovered in children taking only extra dance classes for at least 6 months and children taking no extra classes. The obtained data plays an important role for the design of further investigations of the topic.

**Keywords:** preschool age; executive functions; extracurricular activities; dance classes; preschool education





## 1. Introduction

Executive functions are an umbrella term for cognitive processes that allow for the voluntary control of behavior and cognitive processes and provide purposeful problem solving and adaptive behavior in new situations [1,2]. Executive functions provide the basis for the successful cognitive, emotional, and social development of the child [3]. At the moment, there is an impressive body of evidence confirming their high predictivity in relation to almost all aspects of child development. For instance, the level of development of executive functions in preschool age children significantly determines the quality of children's adaptation to school studies and furthers academic performance [4–7]. The same is true for social competence [8]. Moreover, the level of development of executive functions in children can predict even their socio-economic position in adulthood [9].

Thereby, distinguishing the factors that influence this development is an important task in the context of drawing practical recommendations. At the moment, it is known that one of these factors is the physical activity of a child, for example dance classes [10]. Besides, it is important to explore the relationship between dance classes and executive function development in preschool age children, since this is the period when these functions develop most actively. This is why the main purpose of this work became the investigation of the difference in executive functions in 5–6-year-old children who participated in extracurricular dance classes and children who participated in no such classes.

*1.1. Executive Functions in Preschool Age Children*

We hold to Miyake's model of executive functions [11]. This model was originally based on adult data, but it has successfully been extended to research on the development of executive functions in preschoolers [12]. According to the model, executive functions can be divided into the following components: (1) working memory "involves holding information in mind and mentally working with it" (can be both visual and verbal) [12] (p. 7), (2) cognitive flexibility relates to the ability to switch between rules, tasks, and stimuli, and (3) inhibition supposes the restraint of the dominating response in favor of the one required by the task [11,12].

The main nerve underpinning executive functions is the prefrontal cortex [3]. The maturation of the frontal cortex lasts until early adulthood, respectively, the formation of executive functions lasts the same [3]. The development of the three components of executive functions does not occur simultaneously: rapid development of inhibition control occurs between 2 and 5 years of age [13,14], working memory develops relatively slowly and is closely related to the development of inhibition [15] and cognitive flexibility relies to a great extent on the other two components and develops more slowly [12]. In any case, there is a remarkable boost in the development of executive functions in the senior preschool age [3]. Due to the plasticity of executive functions, their development can be improved through preschool intervention, such as dance classes [3].

*1.2. Dancing and Executive Functions in Children*

A number of research works have demonstrated that taking dance classes could contribute to the improvement of executive functions in children [3,16–19]. For instance, in the study by Shen et al. [3], 4-year-old children that participated in street dance classes for 8 weeks (three times a week, 45 min sessions), demonstrated significantly better performance in a post-test go-no-go task compared to children that participated in no such classes [3]. Lakes et al. [18] organized a 6-week ballet course for 9–14-year-old children with ICP and discovered that the level of inhibitory control of the participants improved. Zinelabidine's and colleague's research also demonstrated an improvement in working memory and cognitive flexibility of 10-year-old children after 2 months of dance classes (twice a week, 45 min sessions) [17]. A randomized-controlled trial conducted by Oppici and colleagues [19] has showed how learning a complex dance choreography for 7 weeks improved 8–10 years old children's working memory capacity. In the study by Suppalarkbunlue W. et al. [14], 4–5-year old children that participated in music–movement training for 8 weeks (45-min, three times a week) showed greater improvements in inhibitory control tasks compared to the control group but showed no significant change in working memory and cognitive flexibility. The study by Rudd and colleagues [16] revealed weak evidence that the choreography classes (twice a week, 8 weeks) improved inhibitory control and working memory more than the creative dance classes (twice a week, 8 weeks) in 6–7-year-old children. However, the authors emphasize that this result may be due to the fact that the creative-dance curriculum was not adopted as planned [16].

Dance classes for children are an organized activity that combines the training of motor skills, aerobic and rhythmic exercises, remembering the sequence of moves, and listening to and understanding music [3]. Thus, dancing combines organized physical activity and musical activity and each of these factors is associated with the improvement of executive functions [3].

Dancing as a form of physical activities stimulates the development of executive functions through a whole variety of physiological mechanisms, such as increasing blood flow, increasing the efficiency of neural networks, increasing the activity of certain brain areas, and general sensorimotor stimulation [20]. From a neuropsychological point of view, organized physical activities (such as dancing, e.g.) and the development of executive functions are related, since both require the activation of the same cerebral zones (cerebellum and prefrontal cortex) [21].

The specifics of dance classes are in the significance of music, which can also influence executive functions. Throughout the whole dance session, children have to be guided by pace and rhythmical structure, the tune, and the mood of the music [3]. This process implies an active participation of sensor, motor, and cognitive systems [3]. Musical activity in preschool age children contributes to the development of various brain structures which, in turn, provide a foundation for cognitive development [22]. Multiple research works revealed that music activities were positively connected to the development of all executive functions' components in children [23–25]. Therefore, it is valid to assume that the musical component of dance classes also plays a significant role from the perspective of executive function improvement. There is also evidence proving that such an improvement can be promoted by training rhythmical skills that are, in fact, required for dancing [3]. Another important role that music plays in dance classes is the harmonization of the emotional environment, reduction of anxiety, and the release of psycho-emotional tension [23]. It allows for the creation of favorable conditions for learning and the development of various cognitive functions of preschoolers [23].

In addition to physical activity and musical activity, there are a number of characteristics inherent in dancing classes, due to which dancing can contribute to the improvement of executive functions. In order to be able to influence executive function improvement, dance training should combine physical challenge and cognitive tasks that become more complicated with time [24,25]. Moreover, in order to improve executive function development, dance classes should be organized targeted at learning complex and diverse movement skills [16,26,27]. Besides cognitive complexity, executive function development requires dance to include motivational elements and meet child's interests and age specifics [25]. In particular, dance classes in preschool age children should not only consist of performing individual exercises and choreography learning but of games, communication with peers and adults, performances, etc. [24].

Dancing involves the active work of all three components of executive functions. During dance class, children are required to control their moves and also inhibit motor impulse not appropriate for the current task. A young dancer has to be able to initiate or stop a required movement in accordance with the choreographer's instructions or the actions of other participants of dancing process. He/she also has to be highly aware of his/her own movements in each particular moment in order to keep up with the correct technique. The results obtained in other research works confirm the connection between organized physical activity and the development of inhibitory control in early ages [25,28]. Any type of dance requires precision for each move and the reproduction of movements in a certain order, in a duration that matches the music, in the right musical moment, and in the right place [3]. Dance classes often take place in front of a mirror and a child has to juxtapose the external image of his/her movement with what the choreographer does and what other children do. Additionally, children need to keep in mind how their movements relate to the music. So, dance classes require an active contribution of working memory. Dance classes involve individual improvisations and spontaneous performances synchronized to music and lesson themes [16,29]. Additionally, during dance classes, one must be ready to react quickly in accordance with changes in movements and changes in the mood and the rhythm of the music [29]. Such types of tasks offer many possibilities for actions and continuous choices, which may challenge inhibitory control and cognitive flexibility [16].

Dance classes require the ability to comply with regimen and rules and to concentrate and distribute one's resources [4]. Therefore, it is also worth considering that the relationship between dance class attendance and the development of executive functions can be bidirectional. It is easier for children with high executive function development levels to comply with the training schedule, follow the instructions of the choreographer, and achieve results in the dance training process.

*1.3. Current Research*

The relationship between dance class attendance in senior preschool age children and executive functions remains insufficiently disclosed and substantiated. This connection is more studied in school students and adults, while there is a lack of evidence for preschool age children [7,30,31]. Moreover, in all the reviewed research works involving children of a preschool age, educational experiments were conducted. Such research designs imply that all participants completed a course of extra classes and they lasted on average 8 weeks. However, in reality, children normally attend extra dance classes longer and often miss classes. Thus, educational experiments have low ecological validity and do not reflect in full the influence of extracurricular dance classes on the development of executive functions in real life.

The current study aimed to investigate the difference in executive functions components in 5–6-year-old children who participated in extracurricular dance classes and children who did not participate in any extracurricular classes. All children from both groups participated in general music, sport, and dance classes offered by kindergartens. In many countries, children attend classes aimed at their cognitive, language, physical, artistic and aesthetical, and social communicative development in the framework of the kindergarten educational program [32]. Dance classes in the framework of the general preschool educational program usually have fixed yearly plans and structures and do not account for individual difference [33]. During such classes, all children usually follow the same instructions and learn through repetitive actions [33]. However, in case of extracurricular dance classes, children usually receive personal feedback and personal instructions. Additionally, children can choose the type of extracurricular dance activities (e.g., ballet, street dance, etc.) according to their individual characteristics and interests. So, extracurricular dance classes may be more effective in executive functions developing then general classes only. The main hypothesis of the research suggested that children attending extra dance classes for at least 6 months had a higher level of executive functions, compared to their peers that participated in no extracurricular classes. This study involved only children that participated in dancing for at least 6 months, because the influence of such classes is more noticeable after 6 months than after short-term courses [3,27].

## 2. Materials and Methods

*2.1. Methods*

2.1.1. Questionnaire for Parents Regarding Dance Class Attendance

We used a questionnaire for parents to find out which extracurricular classes their children attended, for how long, and what their content was. The questionnaire also included questions on other extracurricular classes besides dancing, demographic questions (age and sex of the child), and questions on socioeconomic characteristics of the family (socioeconomic status, mother's education, etc.).

The questionnaire consisted of the following questions related to dancing:

1. Does your child attend dance extracurricular classes? If "yes":
2. What type of dance classes does your child attend?
3. How many times a week does your child attend extracurricular dance classes?
4. What is the average duration of an extracurricular dance class? (min)
5. For how many years has your child been taking these extracurricular dance classes?

2.1.2. Executive Functions Assessment

The main components of executive functions were assessed by means of a set of techniques approbated on and for Russian preschoolers [34,35].

Cognitive inhibition was assessed using the NEPSY-II subtest [36] "Inhibition". This technique consists of a series of 40 figures (squares, circles, and arrows). The tool has two parts: Naming (the child is required to name the figures as quickly as possible) and Inhibition (reverse Naming, i.e., if the child sees a square, he/she should say "a circle", and so on). Time spent on each separate task is registered as well as the number of errors

and the number of self-corrections. These three indicators are translated into a combined scaled score (from 1 to 20 points) in accordance with corresponding tables. The combined scaled score for "Inhibition" was included in the analysis, which accounted for the time and number of errors made while performing the tasks on inhibition.

Physical inhibition was assessed using the NEPSY-II subtest [36] "Statue". In this task, a child has to remain motionless for 75 s, avoiding being distracted by external sound stimulus (knocking, cough, a sound of a pen that fell on the floor, etc.). For each 5–second slot a child received 0–2 points for successfully following the instructions (maximum number of points = 30). Any committed errors such as moves, opening his/her eyes, and sounds were registered as well. The total score was analyzed in the article.

Verbal working memory was assessed using the NEPSY-II subtest [36] "Sentences Repetition". This tool consisted of 17 sentences that gradually became harder to remember because of increasing lengths and more complex grammatical structures. Each sentence correctly repeated was assigned 2 points; if there were one or two errors, they received 1 point; if there were 3 errors or more, 0 points (maximum number of points = 34). In this article, only the total score assigned for the correct repetition of sentences was included in the analysis.

Visual working memory was assessed using the NEPSY-II subtest [36] "Memory for Designs". The children were presented with a picture in which colorful images were located in different cells of a field (four trials with 4, 6, 6, and 8 images). The children were allowed to look at the picture for 10 s, then it was withdrawn, and the respondents had to select the correct images from the set and locate them in corresponding cells on a blank field. Each correctly selected card was assigned 2 points (1 point if a similar card was picked) and 1 point was given to each correctly chosen location. Additionally, 2 bonus points could be received for each complete match with the example (right card in the right place). Afterwards, the content score (max = 48), location score (max = 24), bonus score (max = 48), and total score (max = 120) were calculated. The total score was analyzed in the article.

Cognitive flexibility was assessed by "Dimensional Change Card Sort" task [37]. This tool consisted of three series of tasks where children were to sort cards with the images of rabbits and boats in accordance with different rules. In the first series, 6 cards were to be sorted by color (red were to be put to one side, blue to the other). In the second trial, 6 cards were sorted by shape (boats and rabbits separately). In the third task, children were supposed to be guided by an external stimulus not related to color or shape, i.e., black frame on the picture. The 12 cards were to be sorted depending on the shape or color of the object plus the frame. Each card sorted correctly received 1 point and then the total score was calculated (max = 24). In this article, we used the total score reflecting the number of cards sorted correctly.

*2.2. Procedure*

The assessment of executive functions was carried out individually for each child. The executive functions assessment was performed during two 20 min sessions with a few days break in-between. The tasks were given to all the children in the same order and with the same instructions. Children were tested in their kindergarten in a familiar quiet room (bedroom or psychologist's office). The assessment was performed by specially trained testers. The children were allowed to stop the assessment procedure if for some reason they did not feel like continuing.

Besides the assessment of children, an online questionnaire was taken by their parents. Parents received a link to the survey via email from municipal educational organizations. All parents who participated in the survey also gave informed consent to participate in the study. The approximate time to complete the online questionnaire was 20 min. However, not all the parents who received the link for the questionnaire filled it in. Thus, we can assume that the parents that completed the questionnaire were better organized people with higher involvement in the matters related to the upbringing of their child.

This study and its consent procedures were approved by the Ethics Committee of Faculty of Psychology at Lomonosov Moscow State (approval no. 2021/37). All parents provided written informed consent for their child's participation in the study.

*2.3. Sample*

Typically developing 5–6-year-old children and their parents participated in the study. The children were attending various pre-kindergarten classrooms located in the republics of Tatarstan and Sakha (Yakutia). There were two groups of participants: children only taking up music for at least 6 months ($n = 41$) and children that did not participate in any extracurricular classes ($n = 45$). In each group, about 80–85% were girls, 70% of mothers had higher education, about 80% of families had medium level of income, and approximately 70% of the children were from Yakutia (see Table 1).

**Table 1.** Differences in gender, maternal education, family income, region between groups.

| | | No Extra Classes, $n = 45$ | Taking Extra Dance Classes, $n = 41$ | Differences Chi-Squared Test | *p*-Level |
|---|---|---|---|---|---|
| Gender | Boys | 8 | 6 | 0.156 | 0.693 |
| | Girls | 37 | 35 | | |
| Maternal education | Secondary school | 1 | 2 | 1.095 | 0.895 |
| | Lower post-secondaryVocational education | 9 | 6 | | |
| | Incomplete higher Education | 2 | 3 | | |
| | Higher education | 31 | 29 | | |
| | PhD | 1 | 1 | | |
| Family income | Low | 5 | 2 | 2.161 | 0.339 |
| | Medium | 35 | 31 | | |
| | High | 4 | 7 | | |
| Region | Kazan | 14 | 11 | 0.191 | 0.662 |
| | Yakutia | 31 | 30 | | |

The formation of these two groups developed in several stages (see Figure 1). First, the parents of the 820 children completed the questionnaire online; 154 children attended extracurricular dance classes and 69 of them participated in dancing only and for at least 6 months. The preschoolers involved in extracurricular dance activities for less than 6 months were not included in the sample; 220 of the 820 children whose parents completed the questionnaire did not attend any extracurricular classes. For each respondent taking extra dance classes at least for 6 months, a preschooler of the same age, gender, region, mother's education level, and family income level was selected from the 222 children taking no extra classes. In the end, there were 69 children in each group, 12 male and 57 female participants. This gender distribution inequality in groups is because there were only 12 boys in the "extra dance classes" group. The diagnostics were held for 41 children from the "extra dance classes" group and 45 children from the "no additional classes" group.

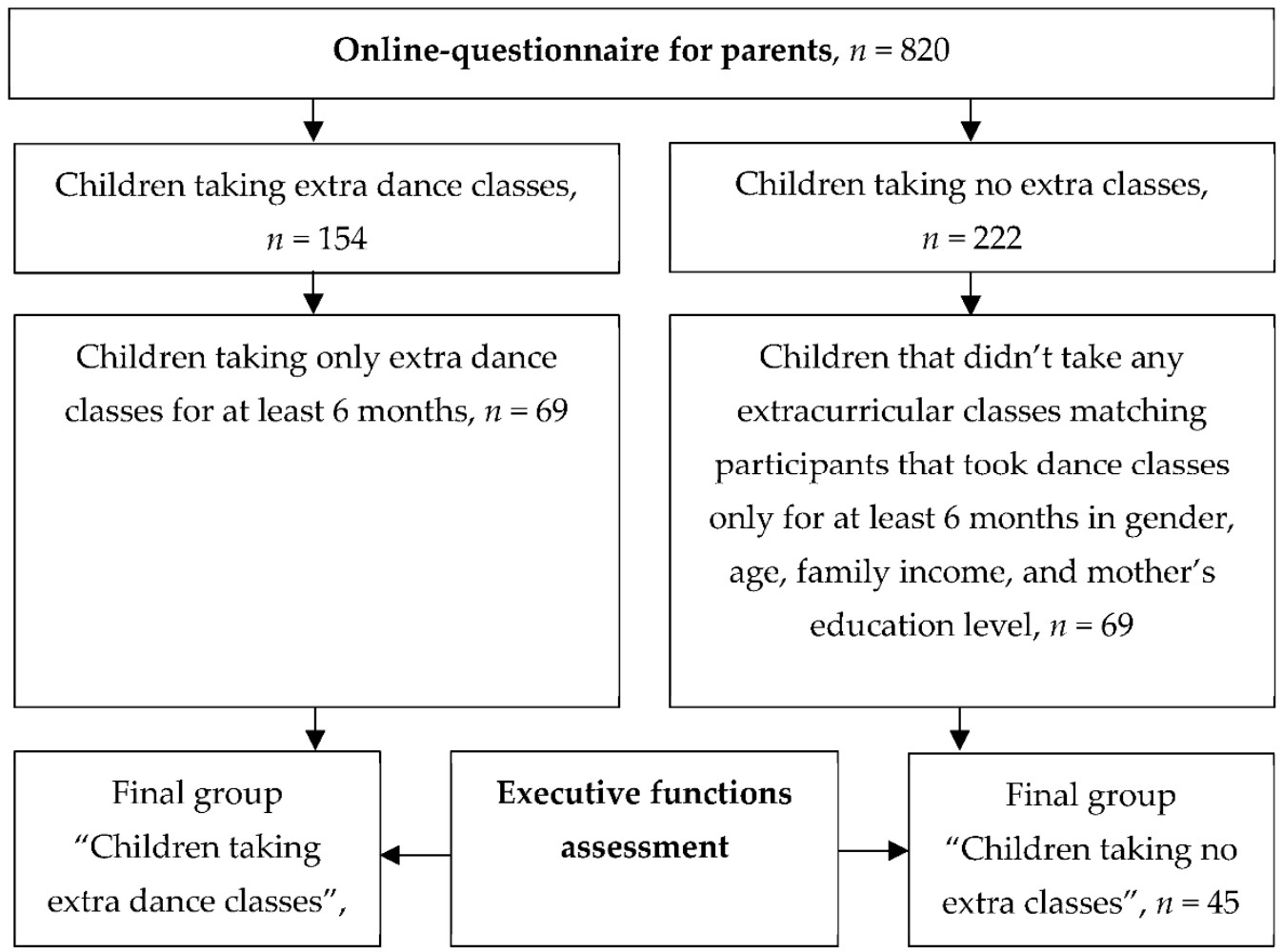

**Figure 1.** Sampling process.

*Group "Children Taking Extra Dance Classes"*

At the moment of filling in the questionnaire, mothers reported that approximately 48% of the children had been taking up extra dance classes for one year, 29% for 2 years, and 23% for 6–11 months. The majority of respondents attended dance classes twice a week (81% for 2 times a week, 8% for 1 time a week, 11% for 3 times a week). Furthermore, 49% of the children from the "extra dance" sample participated in classes of 50–60 min, 37% for 30–40 min, 6% for less than 30 min, and 9% for over an hour.

*Group "Children Taking No Extra Classes"*

These participants, same as the children taking extra dance classes, were following their kindergartens' educational programs [32]. There are musical and rhythmical dance classes twice a week in senior preschool groups (5–6 years old children) in kindergartens.

### 3. Results

Descriptive statistics for all the study variables and ages are presented in Table 2. According to the Kolmogorov–Smirnov test, not all parameters were distributed normally; that is why the nonparametric Mann–Whitney U test for independent samples was used onwards.

**Table 2.** Descriptive statistics for executive functions measures and age.

| | Group | $n$ | Min. | Max. | Mean | Median | SD | Skewness | SD | Kurtosis | SD |
|---|---|---|---|---|---|---|---|---|---|---|---|
| Age (months) | No extra classes | 42 | 54 | 76 | 67.71 | 68.50 | 4.73 | −0.52 | 0.36 | 0.41 | 0.72 |
| | Extra dance classes | 35 | 54 | 78 | 68.54 | 69.00 | 4.97 | −0.79 | 0.40 | 1.48 | 0.78 |
| DCCS | No extra classes | 35 | 16 | 24 | 21.60 | 22.00 | 2.28 | −0.88 | 0.40 | 0.21 | 0.78 |
| | Extra dance classes | 34 | 18 | 24 | 21.56 | 21.50 | 1.97 | −0.22 | 0.40 | −1.05 | 0.79 |
| Inhibition | No extra classes | 45 | 5 | 19 | 10.78 | 10.00 | 3.52 | 0.80 | 0.35 | −0.16 | 0.70 |
| | Extra dance classes | 41 | 6 | 17 | 11.00 | 10.00 | 2.65 | 0.57 | 0.37 | 0.40 | 0.72 |
| Statue | No extra classes | 34 | 16 | 30 | 25.85 | 27.00 | 3.35 | −1.44 | 0.40 | 1.73 | 0.79 |
| | Extra dance classes | 34 | 18 | 30 | 26.15 | 27.00 | 2.60 | −1.28 | 0.40 | 1.96 | 0.79 |
| Memory fordesigns | No extra classes | 34 | 53 | 113 | 78.21 | 74.50 | 17.65 | 0.44 | 0.40 | −0.96 | 0.79 |
| | Extra dance classes | 32 | 54 | 113 | 79.06 | 79.00 | 13.94 | 0.16 | 0.41 | −0.41 | 0.81 |
| Sentence Repetition | No extra classes | 32 | 10 | 28 | 16.97 | 17.00 | 3.76 | 0.93 | 0.41 | 1.91 | 0.91 |
| | Extra dance classes | 32 | 10 | 25 | 15.94 | 15.50 | 2.72 | 1.00 | 0.41 | 3.56 | 0.81 |

According to the Mann–Whitney U test, there are no differences in executive functions levels between the children taking only extra dance classes for at least 6 months and the participants not involved in any extra activities (see Table 3).

**Table 3.** Mann–Whitney U test for differences between mean rank of group taking extra dance classes and group not taking any classes.

| Subtest | No Extra Classes Mean Rank | Extra Dance Classes Mean Rank | Differences Mann–Whitney U Test | $p$-Level |
|---|---|---|---|---|
| DCCS | 35.84 | 34.13 | 565.500 | 0.719 |
| Inhibition | 40.97 | 46.28 | 808.500 | 0.320 |
| Statue | 34.71 | 34.29 | 571.000 | 0.931 |
| Memory for Designs | 32.63 | 34.42 | 514.500 | 0.705 |
| Sentence Repetition | 36.02 | 28.98 | 399.500 | 0.127 |

The group of participants taking extra dance classes included children attending classes for different periods and with different frequencies as well as classes of varying durations. It was possible to unite children attending classes for different periods (from 6 months to 2 years), classes of varying durations (20–90 min), and those attending classes with different frequencies (1–3 times a week) in one group because there was no correlation between the periods when participants participated in dance classes and the level of development of executive functions (see Table 4). Neither was there any correlation between the number of minutes per week that the child spent dancing and the executive functions level (see Table 4).

**Table 4.** Correlations between executive function measures and weekly dance class duration and the duration of dance class attendance.

| | | DCCS | Inhibition | Statue | Memory for Designs | Sentence Repetition |
|---|---|---|---|---|---|---|
| Dance class duration, minutes per week | Spearman's correlation coefficient | −0.001 | 0.085 | 0.011 | −0.003 | −0.085 |
| | significance (two-sided) | 0.994 | 0.439 | 0.921 | 0.978 | 0.466 |
| | $n$ | 78 | 85 | 78 | 76 | 76 |
| The duration of dance class attendance, years | Spearman's correlation coefficient | 0.067 | −0.023 | 0.051 | −0.020 | 0.079 |
| | significance (two-sided) | 0.559 | 0.837 | 0.658 | 0.863 | 0.498 |
| | $n$ | 78 | 85 | 78 | 76 | 76 |

## 4. Discussion

The study aimed to investigate the difference in executive functions in 5–6-year-old children who participated in extra dance classes and children who did not participate in any extra classes. The main hypothesis of the research suggested that children attending extra dance classes for at least 6 months had a higher level of executive functions compared to children who participated in no extra classes. The hypothesis was not confirmed. In the study, no significant differences in the level of executive functions between the group of children taking extra dance classes for at least 6 months and the group of children taking no additional classes were found. The absence of differences in executive function levels between "only extra dancing" and "no extra classes" groups can be explained by certain reasons.

The absence of differences can be related to the fact that the groups were thoroughly equalized in order to avoid statistically significant differences by gender, age, mother's education level, and family income. Before the factors of mother's education level and family income were taken into consideration, in the "only extra dancing" group the executive function levels were higher. However, later it was revealed that in the group of children that did not participate in any extracurricular classes, statistically more participants came from families with lower levels of income and less educated mothers. Therefore, we can assume that the development of executive functions is more affected by family conditions and less by the fact of attending extracurricular dance classes [14].

The second reason of absence of differences between groups may be the key role of kindergartens' educational programs, which children from both groups were following [32]. In senior preschool groups, every day there is one class aimed at speech development, development of mathematical representations, and understanding of the world around. There are also musical and rhythmical dance classes twice a week and painting once. The lack of differences might be related to the fact that all children have quite a busy and action-packed educational program in their kindergartens and, in comparison to it, the contribution of extracurricular classes is of less significance. They do not bring in anything particularly transformational in the sense of executive function development.

The next potential reason for the absence of differences between the groups is the format of extra dance classes for 5–6 year old children. There are no strict rules during the majority of such classes; such classes resemble creative games. For example, a choreography teacher can ask his/her students about the mood of some melody, how animals can be imitated in dancing, and so on. Such free and entertaining dancing can contribute to the harmonization of emotional states and communicative development, as well as to the development of the imagination [38,39]. Meanwhile, the development of executive functions requires a continuous challenge [3,27]. In order to achieve higher levels of executive function development in the learning process, the complexity of tasks should be gradually increased [3,27]. Moreover, in spite of the similarity of dance classes and sports, the advantages for the executive functions provided by sports may not be noticeable in the former case, since the difficulty of motion tasks set for children in dance classes remains unknown. Perhaps, these sessions are rather an artistic activity, not an athletic one.

The absence of differences in executive function levels between groups can be explained by the study limitations. First, in the current research, almost all the parents skipped the question on the type of dance classes their children were taking. Usually in Russia, group dance classes for 5–6-year-old children consist of warming up, creative and active dance-play, learning a simple choreography, and stretching. However, the specific type and content of dance classes in the study remain unknown, while this factor could actually be key for the development of executive functions [3,16].

Secondly, it remains unknown if children actually attended the classes for the full period. They could have skipped a lot and even though officially they have been in the activity for over 6 months, in fact, it could be much less. The factor of the actual frequency of dance class attendance was not taken over the control in the study, which is important

because dancing effects become better a longer more training period and higher training frequency [3,27].

Thirdly, the majority of the samples in both groups were girls from middle and high socioeconomic status families. One can assume that the people from low socioeconomic status families cannot afford to sign their children up for extracurricular dance classes. A higher socioeconomic status of a family is associated with a higher level of executive functions in preschool children [40]. All study participants were normally developing children. Besides, only those children participated in the research that coped successfully with all the executive function assessment tasks. Thereby, missing differences between the groups can be related to the fact that there were no children with a level of executive functions below the age norm. While preschool children with lower executive function levels (normally coming from families with a low socioeconomic status, children with ADHS and developmental delay, and also boys whose executive functions are less developed than in girls at the age of 5–6 years) benefit from dance in relation to executive function development the most [3,27]. However, in the case of 5–6 years old children with normal or high levels of executive functions, the effect of extra dance may be not perceptible.

The next limitation of the study is not a large sample size. So, the absence of differences in executive function levels can be explained by the insufficient sample sizes. Of the 820 children whose parents responded to the questions in the questionnaire, only 69 participants attended dance classes and did it for at least 6 months. Besides, many other children were involved in more than one kind of extracurricular classes, taking painting or music apart from dancing, for example. Last but not least, we did not have any assessment data for all 69 participants, only for 41. Therefore, the sample size was really small.

The last three limitations of the study are not a possible reason for the lack of differences between the groups. These three limitations are important for planning future studies. First, the design leaves an open question about the direction of causal relationships. We cannot exclude the notion that the participants of this study that participated in extra dance classes for at least 6 months, initially had higher executive function levels compared to their peers that were not involved in dance, or the ones that gave up the classes. Secondly, there were in the "extra dance classes" group children with different durations and frequencies of extra dance classes (from 6 months to 2 years, from 20 to 90 min, from 1 to 3 times a week). It was appropriate to mix participants who attended classes for a different time period and with different frequencies because no correlations between the period of time when the children attended extra classes and their durations were registered. Nevertheless, it would be important to investigate in further research the connection between the period of time when children attend extra classes, their frequency, and their duration with executive functions. The last limitation lies in the absence of control over the factor of the specifics of family upbringing and relationships. Apart from extracurricular dance classes, executive functions also influenced the whole environment inhabited by the child [41,42]. It would be valid to assume that parents of the children taking up extra classes are more involved in the upbringing and education of the child than the parents whose children do not participate in any classes. The educational strategies and values of these two groups of parents may also differ.

Based on all the above-mentioned limitations of our study, we can blueprint the prospects of further research work. First, a longitudinal study is being planned in order to follow up on the development of these children and their relationship with extracurricular dance classes. Besides, it would also be interesting to conduct an experimental study and control the initial executive function levels before the dance course. Secondly, in order to compare the specifics of dancing's influence on executive functions, larger samples are needed and a more precise control of the different variables, such as the content, duration, and frequency of the classes. Thirdly, further research would require an analysis of the specifics of upbringing and features of child–parent relationships in families the study participants come from. It would be also interesting to compare dance classes with sport, arts, and music classes, also popular among 5–6-year-olds.



## 5. Conclusions

This study did not reveal any differences in executive function levels between 5–6-year-old children taking extra dance classes for at least 6 months and participants not involved in any extra activities. Possible reasons for the lack of differences are considered in the paper. Further research work is planned based on the limitations of this study. Further research of this topic is relevant not only for psychological community, but also for parents and educators working in preschool institutions. In order to make a design of targeted influence on the executive functions of preschoolers, it is important to explore the different aspects of connection between extracurricular dance classes and executive function improvement.

**Author Contributions:** Conceptualization, A.T. and Y.S.; methodology, D.B. and O.A.; software, D.B.; validation, D.B. and E.C.; formal analysis, E.C.; investigation, E.C., D.B. and M.G.; resources, E.C.; data curation, M.G., D.B. and O.A.; writing—original draft preparation, E.C.; writing—review and editing, D.B., M.G. and O.A.; visualization, E.C.; supervision, A.T., Y.S. and O.A.; project administration, D.B., A.T. and Y.S. All authors have read and agreed to the published version of the manuscript.

**Funding:** This research received no external funding.

**Institutional Review Board Statement:** The study was conducted in accordance with the Declaration of Helsinki, and approved by the by the Ethical Committee of the Department of Psychology, Lomonosov Moscow State University (approval no: 2021/37).

**Informed Consent Statement:** Informed consent was obtained from all subjects involved in the study.

**Data Availability Statement:** The data presented in this study are available on request from the corresponding author.

**Conflicts of Interest:** The authors declare no conflict of interest. The funders had no role in the design of the study; in the collection, analyses, or interpretation of data; in the writing of the manuscript; or in the decision to publish the results.

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
