# Peer review of "The Relationship between Executive Functions and Dance Classes in Preschool Age Children"

_education, doi:10.3390/educsci12110788_

Round 1

Reviewer 1 Report (Previous Reviewer 3)

The paper is much more scientific and developed a lot compared with the previous version.

However, there were no significant differences in executive functions, i recommend doing more analysis, and examining the differences along other variables such as gender, socio-cultural and economic status. 

References in Russian should be changed to English literature. 

Author Response

Point 1: The paper is much more scientific and developed a lot compared with the previous version.

Response 1: Thank you very much for your attention to our article! The improvement of the article was made thanks to your detailed and thoughtful comments! New changes in the article are marked in red.

Point 2: However, there were no significant differences in executive functions, i recommend doing more analysis, and examining the differences along other variables such as gender, socio-cultural and economic status. 

Response 2: We may not have fully understood your recommendation, so we'll provide a few answers:

1) We equalized groups by gender, age, mother’s education level, and family income (see ”2.3. Sample”). So, there is no differences between groups in gender, socio-cultural and economic status. 

2) Unfortunately, it is impossible to compare the level of executive functions in boys and girls, because the size of the groups is insufficient and disproportionate (15 boys and 72 girls, see ”2.3. Sample”). Similarly, it is impossible to compare the groups of children with different level of maternal education (3 children whose mother’s education is only secondary school, 15 - lower post-secondary vocational education, 5 - incomplete higher education, 60 - higher education, 2 - PhD). It is also impossible to compare children from families with different income levels (7 children from families with low-income level, 66 – medium, 11 – high) in terms of executive functions because of the insufficient and disproportionate size of the groups. However, other studies have shown that both gender and socio-cultural and economic status are significant factors in terms of executive functions formation. Thus, it can be assumed that if comparisons were made, we would see differences. 

We would be grateful if you give us some advice how can we examine the differences along other variables such as gender, socio-cultural and economic status. 

Point 3: References in Russian should be changed to English literature.

Response 3: These Russian-language sources are important in the context of this article, so we would prefer not to remove them. In addition, all of these articles have an abstract in English.

Reviewer 2 Report (Previous Reviewer 1)

Dear Author(s),

Thank you for your responses. I appreciate the work you have done to improve the manuscript. I have only minor suggestions as follows:

1.     Please provide information about the theoretical range in cognitive inhibition task (p. 4);

2.     It is customary to present descriptive analyses first, then correlations;

3.     Pease provide skewness and kurtosis data in Table 3;

4.      Please, format the tables properly.

Best wishes,

A.Kamza

Author Response

Point 1: Dear Author(s), Thank you for your responses. I appreciate the work you have done to improve the manuscript. I have only minor suggestions as follows:

Response 1: Thank you very much for your attention to our article! The improvement of the article was made thanks to your detailed and thoughtful comments! New changes in the article are marked in red.

Point 2: Please provide information about the theoretical range in cognitive inhibition task (p. 4); Response 2: We added information about the theoretical range in the cognitive inhibition task (marked red,  p.4, “2.1.2. Executive Functions Assessment”). However, we would be grateful if you give us some advice what else should we write about the theoretical range in cognitive inhibition task.
  Point 3: It is customary to present descriptive analyses first, then correlations; Response 3: Now descriptive analyses are first (table 2) and then goes the table with correlations (table 4).   Point 4: Pease provide skewness and kurtosis data in Table 3; Response 4: We have provided skewness and kurtosis data in Table 2 (marked red).   Point 5: Please, format the tables properly. Response 5: We formatted tables according to the template.

This manuscript is a resubmission of an earlier submission. The following is a list of the peer review reports and author responses from that submission.

Round 1

Reviewer 1 Report

Dear Authors,

I appreciated reading the manuscript. I also appreciate the hard work the authors have put into this. However, I have some concerns and comments.

Title

The word “development” in the title “The Relationship between Sport and Dance Classes, and the Development of Executive Functions in Preschool Age” is quite problematic as it suggests you measured the developmental change. Meanwhile, your study had a correlational cross-sectional design so the wording seems to be somewhat misleading. Maybe you could entitle your manuscript just as “The Relationship between Sport and Dance Classes, and Executive Functions in Preschool Age”?

Introduction section

First, although the introduction is well written, the “1.1. Executive functions” paragraph misses some broader elaboration. EF, as the main construct in your work, should be described more precisely. Please refer briefly to the following issues: What about the development of excecutive functions (especially in preschool years? What have we learned about it so far? What about EF’s neurological underpinnings? What are the factors impacting the development of EF, beyond the sport? What about broader roles of EF (i.e., emotion regulation, self-regulation, learning, etc.)?

Page 1:

·          „Moreover, the level of development of executive functions in children can predicate even their socio-economic position in adult-hood [6].” – it should be „predict”

·          „1) working memory “involves holding information in mind and mentally working with it” – if it is a citation, then it should be marked with proper bibliographical reference.

Page 3:

·          „Due to the organic immaturity of cerebral structures responsible for executive functions, it is groundless to expect an accelerated development of arbitrariness of mental processes in children under 4, based on their involvedness in physical activities.” – I am not sure why do you think so? Could you elaborate on it?

Method section

·          Please, elaborate on the conditions of the assessment situation. Where did it was carried? In a separate room? Were there any distractors? Who assessed the children? Was he/she trained? Were children asked for informed consent?

·          Please, describe more precisely each EF task (what was the child’s task in each one, exactly how many trials there were, not to mention how many trials there were for each level of the task, etc.)

Results section

Unfortunately, I have the most concerns here:

·          There is a lack of descriptive statistics of EF measures (both text and tables) with M, SD, skewness, kurtosis, K-S tests, etc.

·          Please provide the table of correlations between the quantitative measures (EFs, age, etc.)

·          p. 6:Gender distribution in groups wasn’t even: “sport” group there were 15% of girls, and in the “dancing” one, 61%.” – should be „was not equal”

·          par. 3.2. Relationship between Extracurricular Sports Classes and the Development of Executive Functions – did the groups differ significantly in age or sex?

·          par. 3.3. Relationship between Extracurricular Dance Classes and the Development of Executive Functions – please provide detailed data on age (M, SD) and sex of the subgroups; did the groups differ significantly in age or sex?

·          par. 3.4. Comparison of Executive Functions Level in Children Taking Up Extra Sport or Dance Classes Only for at Least a Year - please provide detailed data of age (M, SD) and sex of the subgroups; did the groups differ significantly in age or sex?

·          page 5: „50% of the children were bilingual” – what were the proportions of bilingual children in each subgroup of comparisons („dancers” vs. „no-dancers”, and „sportsmen” vs. „no-sportsmen”)? 

·          There is some empirical evidence that preschool girls outperform boys in executive function tasks (e.g., https://www.tandfonline.com/doi/abs/10.1080/09297049.2013.822060). Furthermore, in preschool years increased development of EF is observed and discussed in broad literature. Moreover, bilingualism and verbal ability in children were found to be linked to their EF (e.g., https://www.frontiersin.org/articles/10.3389/fpsyg.2020.574789/full). All this evidence prompts researchers to control for child age, sex, and verbal abilities in the statistical analyses. Without this, we do not know whether the obtained results are artifacts or not. The observed differences in your study might be significant due to the aforementioned variables. Therefore I strongly recommend repeating your analyses with some more rigorous and advanced statistical methods (e.g. regression analyses), including controlling for age and sex, and bilingualism.

Discussion section

  • p. 9: „At the same time, from the perspective of the cultural-historical approach, it still follows that it is the environment that affects the development of the child [56].” – according to contemporary knowledge, we rather talk about bidirectional influences between the dispositions of the child (with his/her genetics) and the environment (e.g. Lewis, C., Carpendale, J. I. M. (2009). Introduction: Links between social interaction and executive function. In: C. Lewis and J. I. M. Carpendale (red.), Social interaction and the development of executive function. New Directions in Child and Adolescent Development, 123, 1–15.)
  • p. 9, „Therefore, the difference in executive functions level between groups can be related not only to the classes but also to the specifics of family upbringing and relationship.” – there are a lot of works about it so some bibliographic references should be mentioned here (e.g., https://bmcpsychology.biomedcentral.com/articles/10.1186/s40359-021-00524-7)
  • Given the well-documented low commonality of individual EF tasks, the results of the present study might not be generalizable to other EF tasks measuring the same EF component or to other EF components such as working memory and cognitive flexibility. I think this should be highlighted and reflected in the discussion of the results. 

Conclusions section

p. 9 „In order to design of targeted influence on executive functions of preschoolers, (…)” – in order to do sth (instead of: „in order to sth”).

Reviewer 2 Report

“The Relationship between Sport and Dance Classes, and the Development of Executive Functions in Preschool Age”

This paper is interesting as it examines ways to improve children’s executive functions. However, the authors cite a lot of other research on the topic, so might they mention how this research advances the field of study?

And in my opinion, I beleieve there is a weak methodology and interpretation of results.

More specifically:

The introduction should be shortened. There is a lot of information that does not help.

Methodology: The assessment tools for each of the executive function components require further information. What are the children doing in each one so that they can be evaluated?

The readers could be informed about the validity of the NEPSY II test, by authors.

Sample: The children took part in extracurricular activities, sports, and dancing, according to the authors. However, in which games did they participate? How many of them took part in either team or individual sports? What type of dance did they perform? Maybe different kinds of exercise/dance could affect the development of executive functions.

In addition, the authors did not examine whether the children took part in any physical education classes or physical activities at school.

Statistical analysis/results: Although the authors mention in the method section that they will assess the main components of the executive functions, they only interpret the findings of two of these components. Each element of the executive functions must be interpreted, and the findings must be written.

In addition, was there any difference between boys and girls?

Discussion: In this section, authors write a lot of information that would be written in the introduction.

Reviewer 3 Report

The introduction should contain the main goal and research questions of the paper. In the subchapter entitled current research, it can be read that many Russian children take part in extracurricular activities. It is too general . How many? Which part of Russian society? In all social strata in Russia? The relevance of the paper is not established enough scientifically. It is important for Russian researchers who investigate these topics, but why is it interesting other researchers in other countries?

The sampling and the main characteristics and type of the sample (for example RCT or non-randomized, or cluster RCT or other???) is not introduced, however it is essential regarding an intervention program. All the information from the results referring to selection of experimental subjects should be contained by methodology. The characteristics of experimental and control groups should be introduced in much more detail. It is not clear even the numbers of the two groups. 

At the M-W-U test the rank means should be interpreted.

In theoretical background we can read about the difference between open-skilled and close-skilled sports. Why is it not examined in this research by authors? We can see just four analyses, which is a very low number for a scientific study, however there are some other opportunities for comparison (for example gender in both groups or sports etc.), Therefore the analyzes and the results are very poor.